# Large-Dynamic-Range and High-Stability Phase Demodulation Technology for Fiber-Optic Michelson Interferometric Sensors

**DOI:** 10.3390/s22072488

**Published:** 2022-03-24

**Authors:** Wanjin Zhang, Ping Lu, Zhiyuan Qu, Jiangshan Zhang, Qiang Wu, Deming Liu

**Affiliations:** 1Wuhan National Laboratory for Optoelectronics (WNLO) and National Engineering Laboratory for Next Generation Internet Access System, School of Optical and Electronic Information, Huazhong University of Science and Technology, Wuhan 430074, China; zhangwanjin@hust.edu.cn (W.Z.); d201880622@hust.edu.cn (Z.Q.); dmliu@hust.edu.cn (D.L.); 2Huazhong University of Science and Technology Research Institute, Huazhong University of Science and Technology, Shenzhen 518000, China; 3Department of Electronics and Information Engineering, Huazhong University of Science and Technology, Wuhan 430074, China; zhangjs@hust.edu.cn; 4Faculty of Engineering and Environment, Northumbria University, Newcastle upon Tyne NE1 8ST, UK; qiang.wu@northumbria.ac.uk

**Keywords:** phase demodulation, fiber-optic Michelson interferometric sensor, 2 × 2 coupler, coupling ratio, acoustic sensor

## Abstract

A large-dynamic-range and high-stability phase demodulation technology for fiber-optic Michelson interferometric sensors is proposed. This technology utilizes two output signals from a 2 × 2 fiber-optic coupler, the interferometric phase difference of which is π. A linear-fitting trigonometric-identity-transformation differential cross-multiplication (LF-TIT-DCM) algorithm is used to interrogate the phase signal from the two output signals from the coupler. The interferometric phase differences from the two output signals from the 2 × 2 fiber-optic couplers with different coupling ratios are all equal to π, which ensures that the LF-TIT-DCM algorithm can be applied perfectly. A fiber-optic Michelson interferometric acoustic sensor is fabricated, and an acoustic signal testing system is built to prove the proposed phase demodulation technology. Experimental results show that excellent linearity is observed from 0.033 rad to 3.2 rad. Moreover, the influence of laser wavelength and optical power is researched, and variation below 0.47 dB is observed at different sound pressure levels (SPLs). Long-term stability over thirty minutes is tested, and fluctuation is less than 0.36 dB. The proposed phase demodulation technology obtains large dynamic range and high stability at rather low cost.

## 1. Introduction

Fiber-optic Michelson interferometric sensors have been widely researched for monitoring vibration, soundwave, acceleration, liquid refractive index, magnetic field, flow velocity and strain because of their advantages such as high sensitivity, light weight, electromagnetic immunity and capability of multiplexing [1,2,3,4,5,6]. Several demodulation technologies have been put forward to interrogate the phase signal of fiber-optic Michelson interferometric sensors, including phase generate carrier (PGC) demodulation [7,8,9,10,11,12,13,14], 3 × 3 coupler demodulation [15,16,17,18] and spectrum demodulation [19,20]. 

PGC demodulation is widely used for fiber-optic Michelson interferometric sensors because it is easy to achieve, and it shows the advantages of a large dynamic range, high stability and capability of multiplexing. However, several nonlinear problems are introduced such as carrier phase delay (CPD) and accompanied optical intensity modulation (AOIM). Dong et al. [12,13,14] made several efforts to overcome CPD and AOIM through methods such as active laser-wavelength scanning by constant variation of the laser drive temperature, using a fiber delay chain and phase-locked amplifier module and using fast Fourier-transform and look-up table methods to calculate phase demodulation depth. These techniques did solve those problems and increased the complexity and cost greatly. Moreover, the frequency of a demodulated signal was limited by the frequency of the carrier signal since the frequency of carrier signal was much higher than that of the demodulated signal, which made it unsuitable for high-frequency signal demodulation. 

3 × 3 coupler demodulation is also a commonly used technique because of its simple structure and capability of wide-band frequency response. Most traditional 3 × 3 coupler demodulation systems rely on the characteristics of a 3 × 3 coupler, which are that the coupling ratio is 1:1:1, and the interferometric phase difference is 2π/3. Traditional 3 × 3 coupler demodulation becomes invalid when those two characteristics are no longer perfect. Zhang et al. [16] calibrated parameters of the 3 × 3 coupler by the ellipse fitting method, which conducts a frequency modulation on the laser and leads to auxiliary amplitude modulation (AAM). An extra photodetector was used to detect the output of the light source to remove AAM in real time. This increased the complexity and cost of the system as well and was only suitable for an unbalanced Michelson interferometer. Liu et al. [18] used a frequency swept laser source with flat intensity output to obtain parameters of the 3 × 3 coupler. Two spectra with a fixed phase difference could be obtained through linear frequency scanning and then essential parameters of the 3 × 3 coupler could be measured. This required a specific, modulated grating Y-branch tunable laser. 

Spectrum demodulation can also be used as long as the optical phase difference (OPD) is small enough to make it possible for an optical spectrum analyzer (OSA) or spectrum acquisition device to collect its spectrum properly. Zhang et al. [20] utilized a nonzero, padded, fast Fourier-transform with Buneman frequency estimation to demodulate an ultrasensitive fiber-optic Michelson microphone. Limited by the sampling rate of the spectrometer, acoustic signals below 2000 Hz were tested, and a significant sawtooth was observed. 

On the one hand, the optical path difference (OPD) of a fiber-optic Michelson sensor can be precisely controlled to make it possible to collect the spectrum by spectrometer, which limits the demodulation method to be suitable for fiber-optic Michelson sensors with small OPDs. On the other hand, limited by sampling rate of the spectrometer, the demodulation method can only be used for a static signal or low-frequency signal. In our previous work, we proposed a linear-fitting trigonometric-identity-transformation differential cross-multiplied (LF-TIT-DCM) algorithm for extrinsic Fabry–Pérot interferometric (EFPI) sensors [21]. The proposed LF-TIT-DCM algorithm is able to demodulate the phase signal of two interferometric signals, the interferometric phase difference of which is π. Two interferometric signals are obtained from two laser wavelengths, and the wavelength difference is odd times half of the free spectrum range (FSR), which increases the complexity and cost of the system and results in a demodulated phase amplitude that is influenced by wavelength difference. For fiber-optic Michelson interferometric sensors, the interferometric phase difference of the interferometric signals from the two ports of the 2 × 2 coupler is exactly π. This characteristic allows the proposed LF-TIT-DCM algorithm to be used for fiber-optic Michelson interferometric sensors.

In this paper, we put forward a large-dynamic-range and high-stability phase demodulation technology for fiber-optic Michelson interferometric sensors. This technology relies on two output interferometric signals from a 2 × 2 coupler, the interferometric phase difference of which is π. A linear-fitting trigonometric-identity-transformation differential cross-multiplication (LF-TIT-DCM) algorithm is applied to those two interferometric signals to interrogate the phase signal. Interferometric phase differences of the 2 × 2 optical couplers with different coupling ratios are tested to be π, which makes the phase demodulation system immune to the coupling ration. A diaphragm-based fiber-optic Michelson interferometric sensor is fabricated, and an acoustic signal testing system is built to prove the aforementioned technology. Large dynamic range is observed at different sound pressure levels (SPLs), and high stability is achieved, regardless of the influence of laser wavelength, optical power and time.

## 2. Materials and Methods

Typical 2 × 2 coupler phase demodulation system schematic is as shown in Figure 1. Light is emitted from a single-frequency laser (SFL). Emergent light enters a 2 × 2 optical coupler through an optical circulator, which transmits the light from the SFL and one reflected interferometric light from the 2 × 2 optical coupler at the same time. Two arms of the 2 × 2 optical coupler are set as reference arm and sensing arm, respectively, which forms a fiber-optic Michelson interferometric sensor. Two reflected interferometric lights from the optical circulator and the 2 × 2 optical coupler are collected and transformed into voltage signal by photodetectors, respectively. Compared with the dual-wavelength demodulation system in our previous work [21], the 2 × 2 coupler phase demodulation system needs fewer laser and optical devices, which is much simpler and costs less.

Transmission matrix of 2 × 2 optical coupler can be expressed as:(1)$$\left[\begin{array}{cc} \sqrt{1-\varepsilon } & i\sqrt{\varepsilon }\\ i\sqrt{\varepsilon } & \sqrt{1-\varepsilon } \end{array}\right],$$
where *ε* is coupling ratio of the 2 × 2 optical coupler. Transmission matrix of reference arm and sensing arm can be expressed as:(2)$$\left[\begin{array}{cc} r_{r}e^{4\pi nL_{r}/\lambda } & 0\\ 0 & r_{s}e^{4\pi nL_{s}/\lambda } \end{array}\right],$$
where *r_r_* and *r_s_* are reflectivity coefficients of the reference arm and sensing arm, respectively, *L_r_* and *L_s_* are lengths of the reference arm and sensing arm, respectively, *n* is refractive index of optical fiber, and *λ* is laser wavelength of the SFL. Amplitude vector of incident light can be expressed as:(3)$$\left[\begin{array}{c} E_{0}\\ 0 \end{array}\right].$$

By combining Equations (1)–(3), reflective spectra of the MI optical sensor can be calculated as:(4)$$\begin{array}{l} {\left|\frac{E_{1}}{E_{0}}\right|}^{2}=\left[\left({\left(1-\varepsilon \right)}^{2}r_{1}^{2}+\varepsilon^{2}r_{2}^{2})-2r_{1}r_{2}(1-\varepsilon )\varepsilon \cos(4\pi n(L_{1}-L_{2})/\lambda \right)\right]\\ {\left|\frac{E_{2}}{E_{0}}\right|}^{2}=(1-\varepsilon )\varepsilon \left[\left(r_{1}^{2}+r_{2}^{2})-2r_{1}r_{2}\cos(4\pi n(L_{1}-L_{2})/\lambda +\pi \right)\right] \end{array}.$$

In this case, output voltage signals of two photodetectors can be expressed as:(5)$$\begin{array}{l} V_{1}=k_{1}I[({\left(1-\varepsilon \right)}^{2}r_{r}^{2}+\varepsilon^{2}r_{s}^{2})-2r_{r}r_{s}(1-\varepsilon )\varepsilon \cos(4\pi n(L_{r}-L_{s})/\lambda +\varphi (t))]\\ V_{2}=k_{2}I[(1-\varepsilon )\varepsilon (r_{r}^{2}+r_{s}^{2})+2r_{r}r_{s}(1-\varepsilon )\varepsilon \cos(4\pi n(L_{r}-L_{s})/\lambda +\varphi (t))] \end{array},$$
where *k*_1_ and *k*_2_ are related to photoelectric conversion coefficients, *I* is incident intensity and *φ*(*t*) is phase signal introduced by external signal to be measured. From Equation (5), it can be concluded that the interferometric phase difference of those two output signals is always *π*, regardless of coupling ratio and reflectivity. This characteristic ensures the proposed 2 × 2 coupler phase demodulation system will not be influenced by the performance of 2 × 2 optical coupler, which makes the phase demodulation system stable and robust.

Interferometric contrasts of interferometric lights from the 2 × 2 optical coupler can be expressed as:(6)$$\begin{array}{l} B_{1}=\frac{2(1-\varepsilon )\varepsilon r_{r}r_{s}}{{\left(1-\varepsilon \right)}^{2}r_{r}^{2}+\varepsilon^{2}r_{s}^{2}}\\ B_{2}=\frac{2r_{r}r_{s}}{r_{r}^{2}+r_{s}^{2}} \end{array}.$$

Although coupling ratio does not influence interferometric phase difference, it does influence interferometric contrast instead. According to Equation (6), by defining reflectivity ratio *R* = *r_r_*/*r_s_*, the relationship between coupling ratio, reflectivity ratio and interferometric contrasts can be simulated. Simulation results are shown in Figure 2. For *B*_1_, when (1 − *ε*)*εR* equals to 1, it reaches its maximum. This leads to a curving peak in Figure 2a. On the contrary, when *R* equals to 1, *B*_2_ reaches its maximum and is irrelevant with *ε*, which leads to a straight peak. When coupling ratio is 0.5 and reflectivity coefficients of reference arm and sensing arm are the same, *B*_1_ and *B*_2_ maximize and equal to 1. When coupling ratio varies from 0.4 to 0.6 and reflectivity coefficients of reference arm and sensing arm are the same, *B*_1_ increases from 0.92 to 1 then decreases from 1 to 0.92 while *B*_2_ holds to 1. In this case, even when the coupling ratio deviates from 0.5 slightly, a high contrast interferometric signal can still be obtained.

Once a fiber-optic Michelson interferometric sensor is fabricated, interferometric contrasts are fixed since the coupling ratio and reflectivity coefficients are constant. Furthermore, although interferometric contrast is still impacted by the linewidth of the single-frequency laser, the influence is limited for most single-wavelength lasers, the linewidth of which is below several megahertz. To obtain interferometric contrasts of a fiber-optic Michelson interferometric sensor, a large enough phase variation should be applied to the sensor to ensure that interferometric signals reach their maximum value and minimum value. In this case, interferometric contrasts can be calculated according to the definition of the interferometric contrast.

By combining Equations (5) and (6), output voltage signals of two photodetectors can be rewritten as:(7)$$\begin{array}{l} V_{1}=k_{1}I({\left(1-\varepsilon \right)}^{2}r_{r}^{2}+\varepsilon^{2}r_{s}^{2})[1-B_{1}\cos(4\pi n(L_{r}-L_{s})/\lambda +\varphi (t))]\\ V_{2}=k_{2}I(1-\varepsilon )\varepsilon (r_{r}^{2}+r_{s}^{2})[1+B_{2}\cos(4\pi n(L_{r}-L_{s})/\lambda +\varphi (t))] \end{array}$$

In this case, *φ*(*t*) can be calculated by the LF-TIT-DCM algorithm which is proposed in our previous work [21]. The LF algorithm is firstly applied to those two output voltage signals to calculate intercept and slope of the straight line. Then, combined with contrasts of those two interferometric signals, one normalized signal can be obtained. With the TIT algorithm, two normalized quadrature signals with an absolute value sign are calculated from the normalized signals. After removing the absolute value sign from two normalized quadrature signals, the phase signal can be demodulated with the DCM algorithm. The DCM algorithm is achieved through three steps: differential signals of those two normalized signals are calculated firstly. Then, they are multiplied by each of those two normalized signals to obtain the differential signal of the phase signal to be demodulated. Finally, the phase signal to be demodulated is calculated through integrating its differential signal. The detailed calculation process is not repeatedly presented here.

Compared with the dual-wavelength demodulation system in our previous work, interferometric phase difference is not influenced by laser wavelength anymore, which makes the proposed system in this work much more stable and flexible. Compared with PGC demodulation and spectrum demodulation, a phase carrier signal or spectrum acquisition device is not necessary anymore in the proposed phase demodulation system, which makes it suitable for demodulation of a wide frequency band signal. Furthermore, compared with 3 × 3 coupler demodulation, the proposed phase demodulation is hardly influenced by the imperfectness of the 2 × 2 coupler. Through calculating a normalized signal and utilizing the DCM method to complete the phase demodulation, the proposed phase demodulation system is not influenced by the optical power, characteristics of photodetectors and laser wavelength, which makes it rather stable and robust.

## 3. Results and Discussion

### 3.1. Interferometric Phase Difference Experiment

The influence of the coupling ratio of the 2 × 2 coupler on interferometric phase difference was researched. Interferometric signals from four 2 × 2 optical couplers of different coupling ratios were collected, and every two interferometric signals from one 2 × 2 optical coupler were plotted in a coordinate system. Four scatter diagrams are shown in Figure 3, and the coupling ratios in Figure 3a–d are 45:55, 40:60, 35:65 and 30:70, respectively. It is obvious that, no matter how large the coupling ratio is, the scatter diagram is always a straight line, which proves that interferometric phase difference is always π. It is highly consistent with the aforementioned theory.

### 3.2. Acoustic Signal Test

To prove the aforementioned phase demodulation algorithm, a fiber-optic Michelson interferometric sensor was fabricated. A picture of the sensor is shown in the inset of Figure 4. It was mainly comprised of a 2 × 2 optical coupler, a vibrating diaphragm and an external package. The coupling ratio of the 2 × 2 optical coupler was 50:50. Lengths of the reference arm and sensing arm of the 2 × 2 optical coupler were about 120 mm and 140 mm, respectively. The lengths of two arms were restricted to be as short as possible to decrease the influence of polarization-induced fading and make the fiber-optic Michelson interferometric sensor small and compact. Moreover, since the two arms of the fiber-optic Michelson interferometric sensor were closely adjacent to each other, disturbance from environment, such as temperature and vibration, could be regarded as the same for the sensing arm fiber and the reference arm fiber. In this case, the interferometric phase difference of the reference arm fiber and the sensing arm fiber could be rid of environment disturbance. The end faces of the reference arm fiber and sensing arm fiber were both polished to be flat, while a layer of anti-reflection coating was coated on the end face of the sensing arm fiber to make the vibrating diaphragm the only reflecting face in the sensing arm. The diameter and thickness of the vibrating diaphragm were 10 mm and 800 nm, respectively. 

An acoustic signal testing system was built to prove the aforementioned theory and is shown in Figure 4. The light of a single wavelength was emitted from a tunable laser (Alnair Labs, TLG 200, Tokyo, Japan), which replaced the single-frequency laser to research the laser wavelength response to the phase demodulation in the following part. Emitted light entered into a 2 × 2 optical coupler through an optical circulator. Part of the reflected interferometric light was converted to a voltage signal by a photodetector (New Focus, 1623, San Jos, CA, USA) directly, while the other part entered into the optical coupler and was then converted to voltage signal by another PD of the same type. An acoustic signal test was performed with a low-frequency calibration system (Brüel & Kjær, 9757, Nærum, Denmark). The fiber-optic Michelson interferometric sensor was sealed in a low-frequency coupler, where an acoustic signal was generated (Brüel & Kjær, WB-3570, Nærum, Denmark). A signal analyzer (Brüel & Kjær, 3160 PULSE LAN-XI, Nærum, Denmark) was used to output a driving signal, which was amplified by a wide-band amplifier (Brüel & Kjær, WQ-3205, Nærum, Denmark). The two voltage signals from the two photodetectors were collected by the signal analyzer and converted to digital signals. A computer was used to control the signal analyzer and receive digital signals.

An acoustic signal of large amplitude was applied to the fiber-optic Michelson interferometric sensor, and the collected voltage signals and calculated signals are shown in Figure 5. Figure 5a shows the two collected voltage signals from the two photodetectors. Figure 5b shows the scatter diagram of the two collected voltage signals, where a straight line was obtained. After calculating the slope and intercept of the straight line through the linear-fitting algorithm, a normalized signal was calculated and is shown in Figure 5c. Two orthogonal signals with an absolute value sign were calculated from the normalized signal and are shown in Figure 5d. After removing the absolute value sign, the two orthogonal signals were obtained and are shown in Figure 5e. Finally, a phase signal, the frequency of which was 250 Hz, was calculated with the DCM method and is shown in Figure 5f.

### 3.3. Linear Response

The linear response of the proposed phase demodulation system was researched. Acoustic signals of different amplitudes were applied to the fiber-optic Michelson interferometric sensor, and the demodulated phase signals are shown in Figure 6. The proposed LF-TIT-DCM-based phase demodulation algorithm worked properly at different sound pressure amplitudes. Furthermore, more acoustic signals of different amplitudes were applied, and the relationship between the applied sound pressure and demodulated phase amplitude and their linear fitting is shown in Figure 7. When the applied sound pressure varied from 0.024 Pa to 2.86 Pa, the demodulated phase amplitude varied from 0.033 rad to 3.18 rad. More experiments were performed when the applied sound pressure was small to explore the ability of the proposed phase demodulation algorithm to interrogate small signals. The slope of the linear-fitting curve was calculated to be 1.11, which means the sensitivity of the fiber-optic Michelson interferometric sensor was 1.11 rad/Pa. The correlation coefficient between applied sound pressure and demodulated phase amplitude was calculated to be 0.9999, which proves that excellent linear response and large dynamic range are obtained in the proposed phase demodulation system.

### 3.4. Laser Wavelength Response

To prove that the proposed phase demodulation is highly stable, the influence of laser wavelength was researched. The relationship between laser wavelength and demodulated phase amplitude is shown in Figure 8. Experiments were performed at two different SPLs, which were 97.1 dB and 87.5 dB, respectively. When the laser wavelength varied from 1550 nm to 1555 nm, the variations of the demodulated phase amplitude at the two different SPLs were 0.13 dB and 0.47 dB, respectively. Laser wavelength influenced the two collected voltage signals through changing the initial phase of the two interferometric signals. Due to utilization of the DCM algorithm, the demodulated phase amplitude was not influenced by the initial phase of the two interferometric signals. It proves that the demodulated phase amplitude was not influenced by laser wavelength, which is consistent with the aforementioned theory. Moreover, the proposed demodulation system performed better than that of the dual-wavelength demodulation system, the demodulated phase amplitude of which was influenced by laser wavelength since the interferometric phase difference was influenced by wavelength difference.

### 3.5. Optical Power Response

Furthermore, the influence of optical power was also researched. The relationship between optical power and demodulated phase amplitude is shown in Figure 9. Experiments were performed at two different SPLs, which were 97.1 dB and 87.5 dB, respectively. When the optical power varied from 10 mW to 30 mW, the variations of the demodulated phase amplitude at the two different SPLs were 0.13 dB and 0.33 dB, respectively. Although the amplitudes of the two collected voltage signals were influenced by the optical power, owing to the process of linear fitting and normalization, the demodulated phase amplitude was irrelevant to the optical power.

### 3.6. Repeatability Response

The stability of the proposed phase demodulation system over thirty minutes was tested. Experiments were performed at two different SPLs, which were 91.1 dB and 81.4 dB, respectively, and the results are shown in Figure 10. The variation of the demodulated phase amplitude over thirty minutes at the two different SPLs were 0.36 dB and 0.31 dB, respectively. Experimental results show that the proposed LF-TIT-DCM-based 2 × 2 coupler phase demodulation system exhibits high stability over time.

### 3.7. Indoor and Outdoor Tests Comparison

Finally, indoor and outdoor tests were performed to prove the stability of the proposed phase demodulation system further. The whole system was placed near a road to introduce external vibration and noise. All of the optical devices and equipment were placed on the ground directly. A loudspeaker was used as the sound source instead of the low-frequency coupler to expose the fiber-optic Michelson sensor to the environment. The two interferometric signals collected outdoors are shown in Figure 11a, while the two interferometric signals collected indoors are shown in Figure 11b. The signals collected outdoors were much more unstable compared with the signals collected indoors. The demodulated phase signals were as shown Figure 11c, and their frequency spectra are shown in Figure 11d. Although the outdoor noise was 20 dB higher than the indoor noise, the phase signals could still be demodulated.

## 4. Conclusions

In summary, a large-dynamic-range and high-stability phase demodulation technology for fiber-optic Michelson interferometric sensors is proposed. Two interferometric signals, the interferometric phase difference of which is π, are obtained from a fiber-optical coupler. Theoretical analysis and experimental results showed that the interferometric phase difference is not influenced by the coupling ratio and reflectivity of sensing arm and reference arm, which makes the proposed phase demodulation technology robust and stable. The proposed phase demodulation technology mainly depends on a LF-TIT-DCM algorithm to interrogate the phase signal from those two interferometric signals. A fiber-optic Michelson interferometric acoustic sensor was fabricated, and an acoustic signal testing system was built to prove the aforementioned algorithm. Acoustic signals with different sound pressures were applied to the acoustic sensors, and a linear response from 0.033 rad to 3.18 rad was obtained, and the correlation coefficient was as high as 0.9999. Excellent stability was observed, regardless of laser wavelength, optical power and time at different sound pressure. Maximal variation was below 0.47 dB.

## Figures and Tables

**Figure 1 sensors-22-02488-f001:**
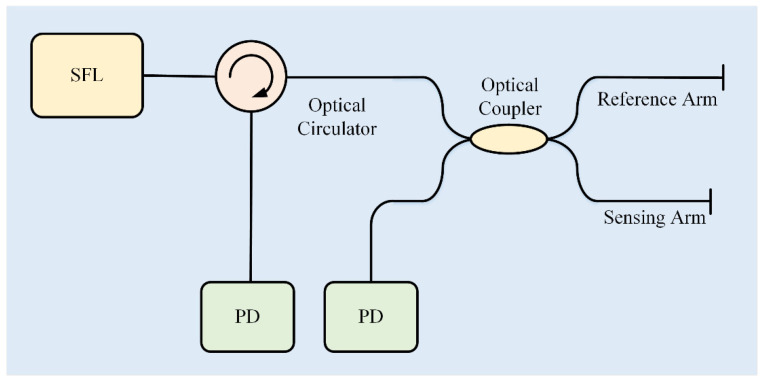
System schematic of 2 × 2 coupler phase demodulation. SFL: single-frequency laser, PD: photodetector.

**Figure 2 sensors-22-02488-f002:**
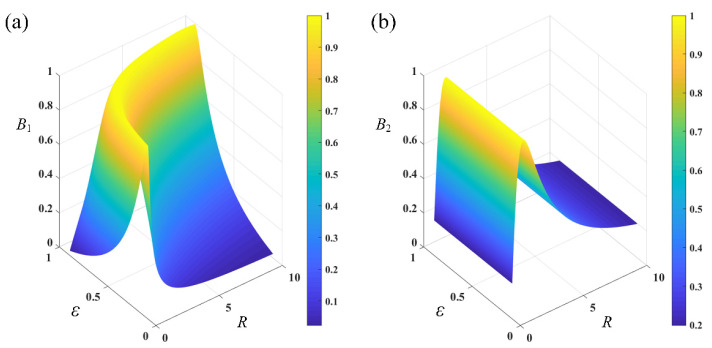
Relationship between coupling ration *ε*, reflectivity ratio *R* and interferometric contrasts *B*_1_ and *B*_2_. (**a**) *B*_1_. (**b**) *B*_2_.

**Figure 3 sensors-22-02488-f003:**
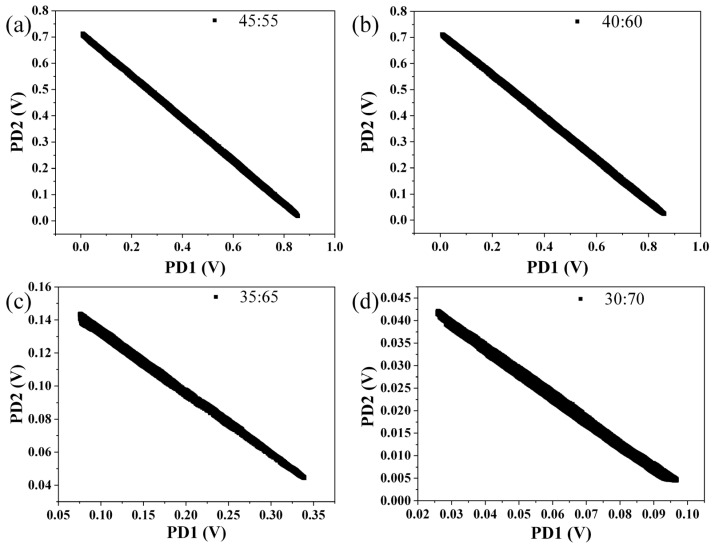
Interferometric phase differences of optical couplers of different coupling ratio. (**a**) 45:55. (**b**) 40:60. (**c**) 35:65. (**d**) 30:70.

**Figure 4 sensors-22-02488-f004:**
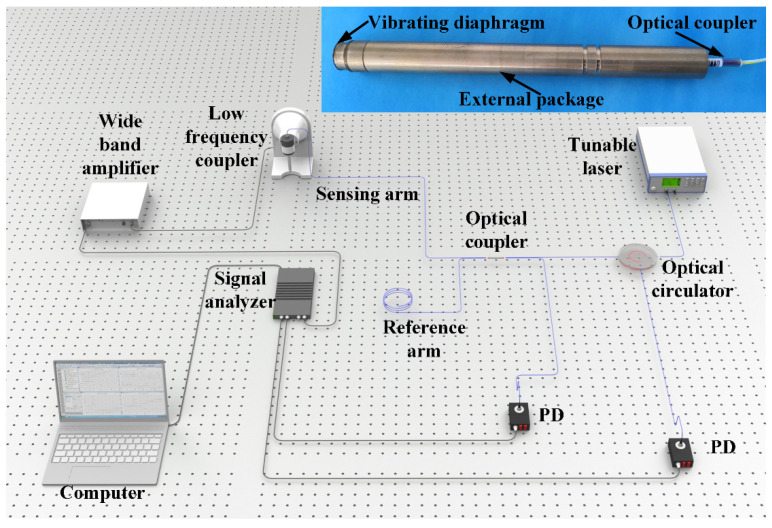
Acoustic signal testing system. Inset: picture of fiber-optic Michelson interferometric sensor. PD: photodetector.

**Figure 5 sensors-22-02488-f005:**
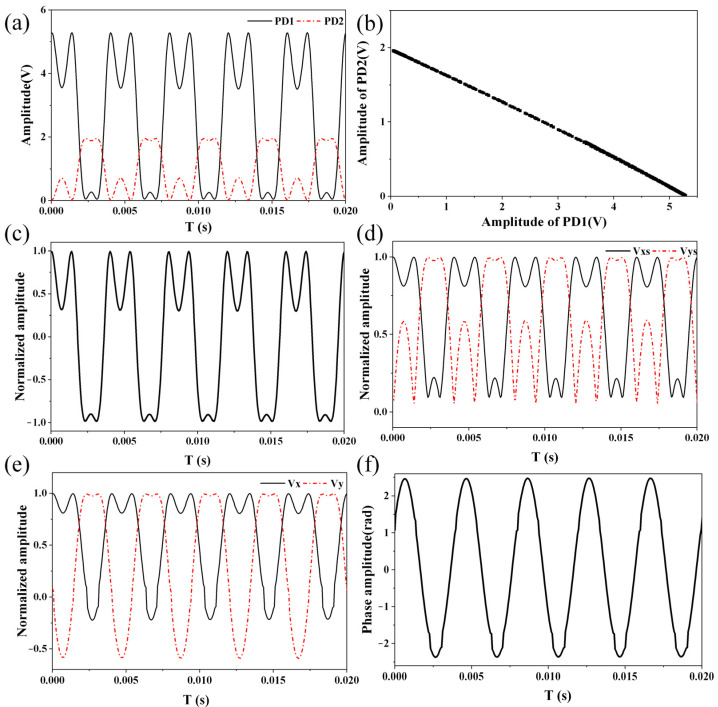
Collected voltage signals and calculated signals. (**a**) Collected voltage signals from two photodetectors. (**b**) Scatter diagram of two collected voltage signals. (**c**) Normalized signal. (**d**) Orthogonal signals before removing absolute value sign. (**e**) Orthogonal signals after removing absolute value sign. (**f**) Demodulated phase signal, the frequency of which was 250 Hz.

**Figure 6 sensors-22-02488-f006:**
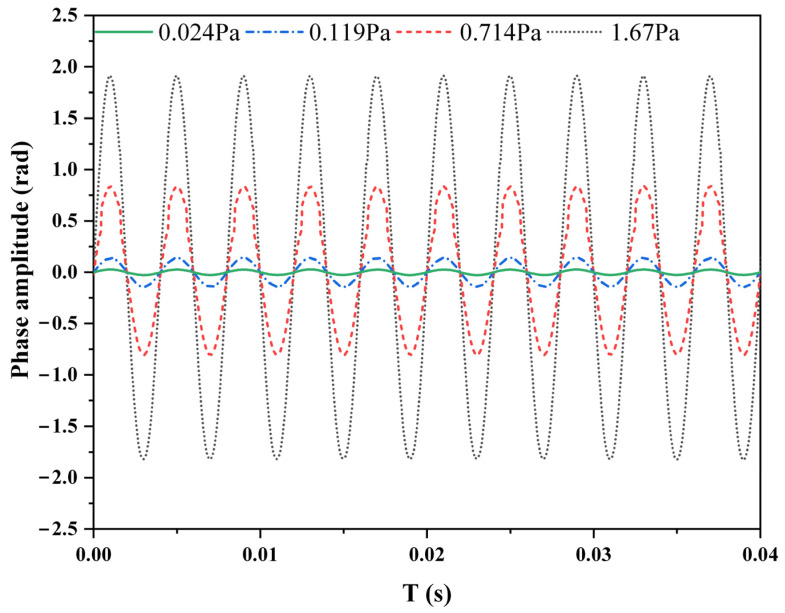
Demodulated phase signal of different amplitudes.

**Figure 7 sensors-22-02488-f007:**
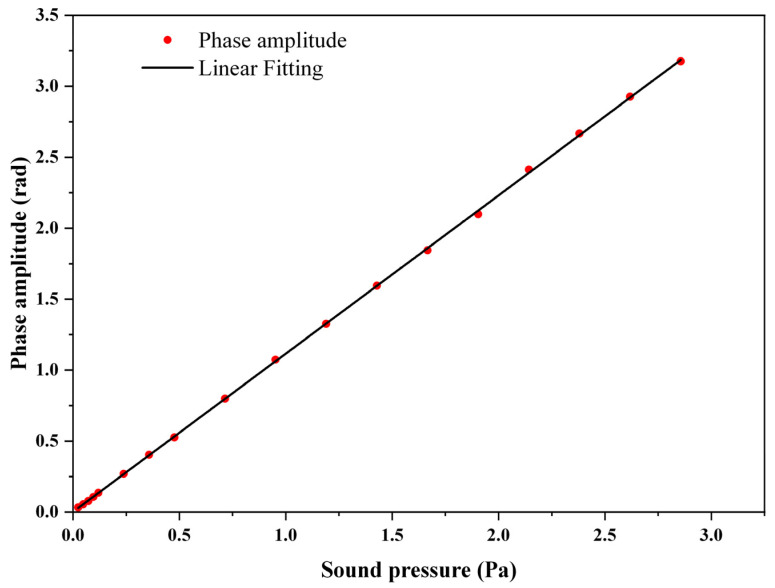
Relationship between applied sound pressure and demodulated phase amplitude and their linear fitting.

**Figure 8 sensors-22-02488-f008:**
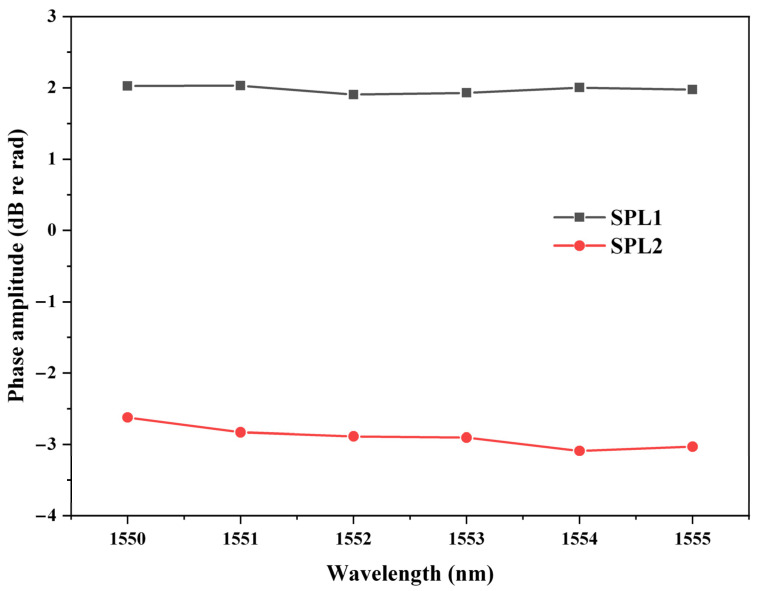
Relationship between laser wavelength and demodulated phase amplitude. SPL1: 97.1 dB, SPL2: 87.5 dB.

**Figure 9 sensors-22-02488-f009:**
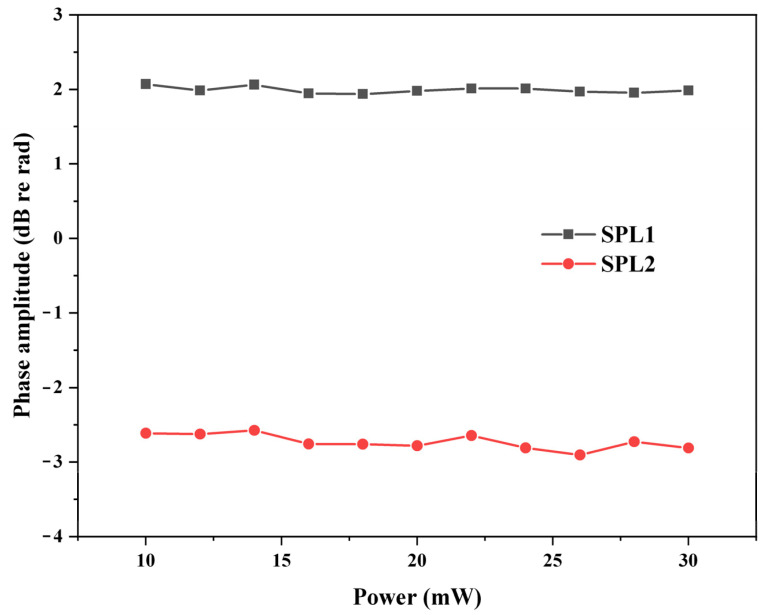
Relationship between optical power and demodulated phase amplitude. SPL1: 97.1 dB, SPL2: 87.5 dB.

**Figure 10 sensors-22-02488-f010:**
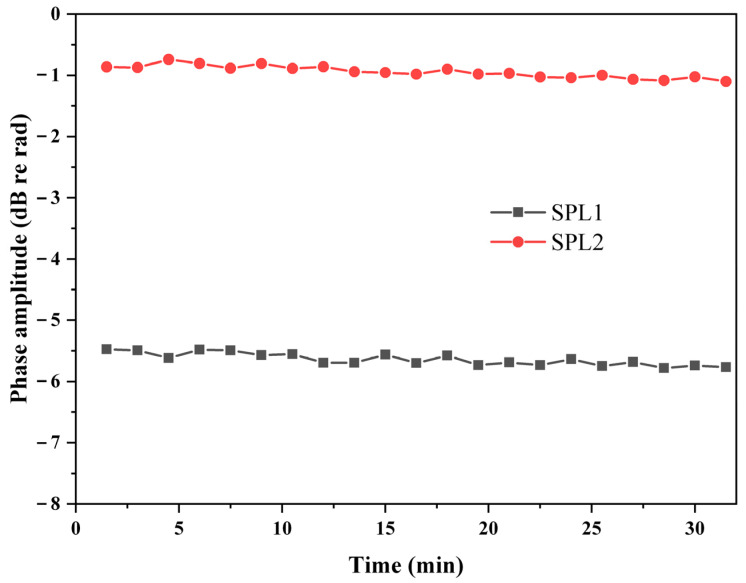
Demodulated phase amplitude over thirty minutes. SPL1: 91.1 dB, SPL2: 81.4 dB.

**Figure 11 sensors-22-02488-f011:**
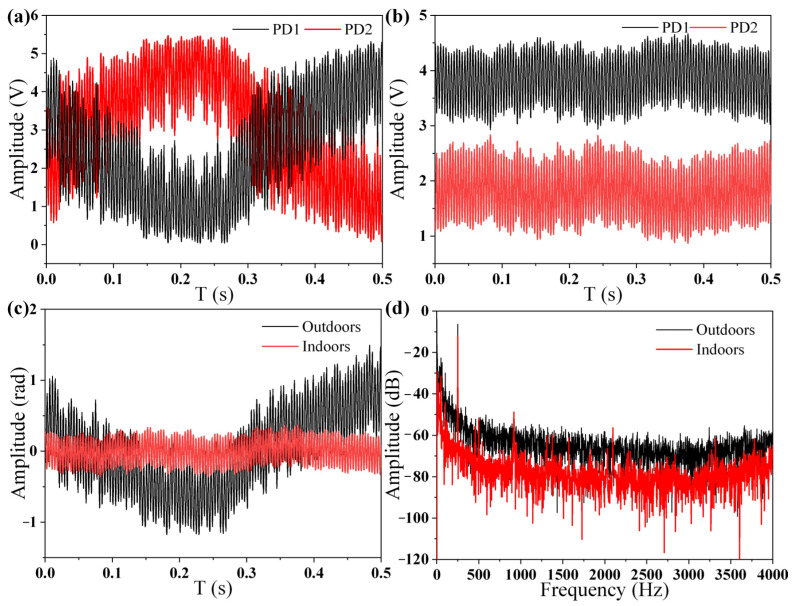
(**a**) Collected signals outdoors. (**b**) Collected signals indoors. (**c**) Demodulated phase signals outdoors and indoors. (**d**) Frequency spectra of phase signals outdoors and indoors.

## Data Availability

The data that support the findings of this study are available from the corresponding author upon reasonable request.

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
