# Peer review of "Large-Dynamic-Range and High-Stability Phase Demodulation Technology for Fiber-Optic Michelson Interferometric Sensors"

_sensors, 2022, doi:10.3390/s22072488_

Round 1

Reviewer 1 Report

  1. Linearity in phase amplitude varies for sound pressure range from 0.024 to 2.86 P. Whether proposed model valid for high pressure range? What is the maximum range of sound pressure can the proposed model support?
  2. Authors are advised to compare the proposed model with recent published all optical sensor technology and mention the advantages of the proposed model.

Reviewer 2 Report

This paper is well written, and describes improvements in a passive homodyne phase demodulation technique based on a linear fitting-trigonometric identity transformation-differential cross multiply (LF-TIT-DCM) algorithm previously reported by the authors for a Fabry-Perot sensor [21]. In this report, the authors achieved, for a fiber optic Michelson interferometric sensor, a larger dynamic range, and higher stability phase demodulation algorithm compared to the results obtained before for a Fabry-Perot sensor. A fiber optic Michelson interferometric acoustic sensor was fabricated and an acoustic signal testing system was built in to prove algorithm performance. This LF-TIT-DCM algorithm is applied to the interferometric signals to interrogate phase changes created by the acoustic signal testing system.

The authors claim that time changes in phase φ(t) can be calculated by LF-TIT-DCM algorithm as proposed in their previous work [21], and the detail calculating process will not be repeatedly presented. After reading their previously work, my opinion is that the DCM algorithm was not clearly presented nor discussed in [21]. So, would be appropriated to add a new paragraph clarifying how the proposed LF-TIT-DCM-based passive homodyne algorithm can demodulate the signal, and how fast it can process the information. 

Furthermore, the fiber optic Michelson interferometer is sealed in a low frequency coupler, where acoustic signal is generated (Brüel & Kjær, WB-3570, Nærum, Denmark). Technical specifications of this sensing element are not reported and could not be found at the supplier. The authors might supply specs to this acoustic source. https://www.bksv.com/en/calibration/calibration-systems/microphone-calibration-systems/low-frequency-calibration-system.

Finally, would be nice to readers if  the authors could add  an extra 3.5a. Section - Acoustic frequency response.

More comments, and suggestions are in the attached pdf file

Reviewer 3 Report

The authors build on their previous work and present another version of their interferometric fibre-optic sensor. For example, the setup looks very similar to that presented in [16], except that the 3x3 coupler has been replaced with a 2x2 coupler and a circulator; one can question whether this makes the setup fundamentally different. In [15], a 3x3 coupler and a circulator is used.  On the other hand, I agree that the linearity shown in fig. 7 is very nice, although the raw data shown in fig. 6 looks very similar to the raw data of fig. 8 in [16]. If they had explored the lower amplitude values in [16], then my assumption is that a similar plot as fig. 7 here could already have been plotted then.

In short, the authors need to clearly articulate the novelty and advantage of the results shown here compared to their previous work.

Secondly, they should advance their work and test it on real applications outside the laboratory in order to support their claim that the "technology is robust and stable". Almost every technology can be made "robust and stable" in the laboratory.  

Thirdly, I notice that most of the applications-based references are dated, and that most of the technology-based references are from a very limited number of groups, with many self-citations. This is not acceptable. Fibre-optic sensors are a mature technology and widely used in industry, so it would be more important to learn about the real issues and how they can be addressed. Please provide a more balanced reference-list and put the work better into context with applications and progress made internationally. 

Round 2

Reviewer 3 Report

I thank the authors for taking my comments into account and particularly for providing outdoor measurememnts, which I agree make the manuscript much stronger.